# From 2D Myotube Cultures to 3D Engineered Skeletal Muscle Constructs: A Comprehensive Review of In Vitro Skeletal Muscle Models and Disease Modeling Applications

**DOI:** 10.3390/cells14120882

**Published:** 2025-06-11

**Authors:** Tianxin Cao, Curtis R. Warren

**Affiliations:** Cardiovascular-Renal-Metabolic Diseases Research Department, Boehringer Ingelheim Pharmaceuticals, Inc., Ridgefield, CT 06877, USA; tianxin.cao@boehringer-ingelheim.com

**Keywords:** 3D engineered muscle tissue, skeletal muscle models, disease modeling

## Abstract

In recent years, the field of skeletal muscle tissue engineering has experienced significant advancements, evolving from traditional two-dimensional (2D) cell cultures to increasingly sophisticated three-dimensional (3D) engineered constructs. While 2D models have provided foundational insights into muscle cell biology, emerging 3D platforms aim to better recapitulate the complex native muscle environment, including mature muscle fibers, supportive vasculature, and native-like extracellular matrix (ECM) composition. Here, we provide a comprehensive review of current in vitro skeletal muscle models, detailing their design principles, structure, and functionalities as well as the advantages and limitations inherent to each approach. We put a special emphasis on 3D engineered muscle tissues (EMTs) developed through advanced bioengineering strategies and note that design criteria such as scaffold selection, perfusion system incorporation, and co-culture with supporting cell types have significantly enhanced tissue maturity and complexity. Lastly, we explore the application of these engineered models to disease studies, highlighting models of both mendelian muscle disorders and common polygenic diseases and the potential of these platforms for drug discovery and regenerative therapies. Although an ideal in vitro model that fully recapitulates native muscular architecture, vascularization, and ECM complexity is yet to be realized, we identify current challenges and propose future directions for advancing these bioengineered systems. By integrating fundamental design criteria with emerging technologies, this review provides a roadmap for next-generation skeletal muscle models poised to deepen our understanding of muscle biology and accelerate therapeutic innovation.

## 1. Introduction

Skeletal muscle is a highly organized tissue composed of multinucleated muscle fibers bundled into fascicles and wrapped in three layers of connective tissue: the endomysium, perimysium, and epimysium. Each muscle fiber is a single elongated cell formed by the fusion of activated myoblasts during development, containing dozens to hundreds of peripherally located nuclei and a densely packed array of centrally located contractile myofibrils [1]. The myofibrils are made up of tandem repeats of sarcomeres, which are the basic contractile unit in muscle [2]. The sarcomeres consist of overlapping myosin thick filaments and actin thin filaments, interconnected by critical structural elements such as the Z-disc, a lattice composed of key structural proteins including α-actinin, titin, and nebulin, which anchor the cytoskeletal networks and stabilize the sarcomere architecture during contraction. This hierarchical structure enables coordinated force generation, originating from neuromuscular junction-controlled molecular cross-bridge cycling within sarcomeres and translating into macroscopic muscle contraction that produces movement of organisms [2,3].

In addition to force production, skeletal muscle serves critical roles in metabolism and physiology. It is a major site of glucose uptake and fatty acid oxidation during exercise and functions as an endocrine organ by secreting myokines during and following contraction, which mediate post-exercise adaptations such as metabolic remodeling, tissue repair, and neuromuscular plasticity [4,5]. Several key physiological factors regulate muscle growth, maturation, and regeneration. Neuronal innervation plays an essential role in this process, as each skeletal muscle fiber requires innervation by an α-motor neuron to form a functional motor unit [6]. Neural activity acts as a key determinant in fiber-type specification and maintenance, influencing both metabolic and contractile properties [7,8,9]. Disruption of this neuromuscular interaction compromises structural and functional integrity for skeletal muscle homeostasis. Mechanical loading is another potent regulator, which promotes muscle hypertrophy due to the adaptation to increased load (e.g., resistance exercise). Chronic mechanical loading under both in vivo and in vitro conditions induces anabolic signaling pathways, such as Akt/mTOR activation, which increase rates of protein synthesis and drive the hypertrophic growth of muscle fibers [10,11]. Skeletal muscle mass is also dynamically regulated by endocrine and paracrine signaling mechanisms. Growth factors, including insulin-like growth factor-1 (IGF-1), fibroblast growth factor-2 (FGF-2), and hepatocyte growth factor (HGF), are secreted by skeletal muscle tissue, motor neurons, and immune cells in response to mechanical stress (e.g., exercise) or injury. These ligands activate pro-myogenic pathways, such as PI3K/Akt and MAPK signaling, to promote satellite cell proliferation, protein synthesis, and muscle fiber hypertrophy [12,13]. Vascularization is another important factor in regulating muscle growth, metabolic homeostasis, and regenerative capacity. In skeletal muscle, the vasculature is characterized by an extensive capillary network aligned longitudinally with muscle fibers. This microvascular architecture facilitates the delivery of oxygen and metabolic substrates while clearing catabolic byproducts, thereby maintaining tissue homeostasis [14].

Historically, skeletal muscle research relied heavily on in vivo animal models and in vitro 2D cell cultures of muscle cells. Conventional 2D cultures typically involve plating myogenic cells on flat plastic or glass surfaces and inducing them to fuse into multinucleated myotubes in a monolayer (Figure 1). Such 2D systems are simple and reproducible, and they have been instrumental in elucidating molecular pathways of muscle gene regulation, protein synthesis, and metabolic responses [15]. Although they have been used to generate critical insights, 2D myotube culture lack 3D tissue architecture and measurable contractile forces, limiting their utility in studying mechanical output or disease-related phenotypes.

To address the limitations of 2D cultures, 3D engineered muscle tissues (EMTs) have emerged as advanced in vitro platforms that better replicate native skeletal muscle structure and function (Figure 1). Here, we define EMTs as 3D constructs combining myogenic cells (e.g., primary myoblasts or iPSC-derived muscle progenitors) with biomaterial scaffolds or self-assembled matrices, designed to mimic the spatial organization, multicellular interactions, and contractile properties of native skeletal muscle. These systems facilitate the culture of muscle cells within scaffold-supported or self-assembled microenvironments, enabling critical processes such as cell alignment, structural maturation, and synchronized contraction under physiological stimuli [16,17,18,19,20,21,22].

This review provides an extensive overview of EMT research from the past two decades. We review in detail the bioengineering strategies employed to create 3D skeletal muscle constructs, including scaffold materials and microarchitecture design, perfusion bioreactors for nutrient delivery, the incorporation of motor neuron inputs to form neuromuscular junctions, co-culture with supporting cell types, and the application of electrical, mechanical, and biochemical stimuli to drive maturation. Next, we summarize how these 3D skeletal muscle platforms are being used to model muscle diseases: from monogenic muscular dystrophies to aging-related sarcopenia and metabolic myopathies. By integrating findings across a broad range of studies, this review aims to identify the key advances and remaining hurdles on the path toward physiologically robust and clinically relevant muscle models.

## 2. Overview of In Vitro Skeletal Muscle Models

### 2.1. Conventional 2D Cell Culture

Two-dimensional cultures of myogenic cells have been used extensively for skeletal muscle biology studies. The most widely used is the C2C12 cell line, a mouse myoblast line derived from adult muscle satellite cells that was selected for robust proliferation and differentiation in vitro [15]. When cultured in high-serum growth medium, C2C12 cells proliferate as mononuclear myoblasts, while switching to low-serum differentiation medium induces cell-cycle exit and fusion into multinucleated myotubes within 72 h [23]. Other rodent cell lines, such as the L6 and L8 rat myoblasts, similarly enable studies of muscle-specific signaling pathways [24]. However, those immortalized rodent cell lines exhibit critical limitations; they are only able to differentiate into fast-twitch type II fibers predominantly, and phenotypic drift occurs within and between different laboratories [25,26].

Human immortalized myogenic cell lines, such as rhabdomyosarcoma-derived cells, which originate from primitive mesenchymal progenitors, retain partial capacity to differentiate into myotubes in vitro. However, they often demonstrate compromised terminal differentiation and aberrant regulation of transcriptional regulators essential for myogenesis [27,28,29]. To overcome this, hTERT (telomerase) and CDK4 (cyclin-dependent kinase 4) have been co-expressed in primary human satellite cells to extend their proliferative capacity while preserving differentiation potential [30]. The resulting immortalized lines, such as LHCN-M2, retain transcriptomic fidelity to primary human myoblasts, including expression of *PAX7* (a satellite cell marker) and *MYOD1* (a myogenic regulator), and form striated, contractile myotubes in vitro when cultured in differentiation conditions [31].

Primary myogenic cultures derived from skeletal muscle biopsies can be a more physiologically relevant model compared to immortalized cell lines. Satellite cells, the resident muscle stem cells, are isolated through enzymatic dissociation using collagenase/dispase followed by cell sorting using surface markers such as CD56/NCAM (human) or α7-integrin (mouse) [32,33]. In culture, these cells activate myogenic programs, upregulating the transcription factors *MYOD1* and *MYF5*, and undergo limited expansion (5–10 population doublings) before terminal differentiation into multinucleated myotubes under low-serum conditions [34]. Primary myotubes exhibit species- and fiber-type-specific characteristics, including expression of adult myosin heavy-chain isoforms (MYH I, MYH IIa, and MHC IIx in humans).

Additionally, induced pluripotent stem cells (iPSCs) have become a key resource for generating human skeletal muscle cells in vitro, offering a renewable and patient-specific platform for disease modeling and therapeutic research. Human iPSCs, reprogrammed from somatic cells (e.g., dermal fibroblasts or peripheral blood mononuclear cells), can be differentiated into skeletal myocytes via directed differentiation and transcription factor-based programming. In transcription factor-based programming, iPSCs are directly converted into myogenic progenitors or myotubes by overexpressing myogenic master regulators such as MYOD1 or PAX7, thereby bypassing early developmental stages and rapidly initiating the myogenic gene network. For instance, Uchimura et al. demonstrated that transient MYOD1 overexpression in iPSCs, coupled with a Matrigel-based culture, rapidly generates myogenin-positive myoblasts capable of forming multinucleated myotubes for drug screening [35]. Although genetic programming can yield myotubes more quickly, it may produce immature fibers and does not fully replicate natural development. On the other hand, directed differentiation guides iPSCs through a stepwise developmental process by treating them with precisely timed signaling factors (e.g., Wnt activation and BMP inhibition) that mimic the molecular environment of embryonic myogenesis [36].

### 2.2. Transition to 3D Models: Bioengineering Strategies for 3D Skeletal Muscle Models

To overcome the deficiencies of 2D cultures, 3D skeletal muscle models can provide a native muscle-like physical and biochemical microenvironment in order to promote structural and functional maturation. In 3D scaffolds, myoblasts align into parallel, striated myotubes resembling in vivo skeletal muscle fibers, driven by ECM-mimetic biomaterials (e.g., fibrin and collagen) and mechanical tension [37,38]. This mechanical support prevents myotube detachment and atrophy, enabling sustained contractions and ultrastructural maturation (e.g., T-tubule formation and sarcoplasmic reticulum development) [39]. Three-dimensional systems also facilitate multicellular integration (e.g., endothelial cells and motor neurons) to mimic skeletal muscle’s native niche. Co-culture with motor neurons induces functional neuromuscular junctions, evoking contraction upon neuronal stimulation [40,41]. The 3D environment promotes proper cell alignment, differentiation, and functional assembly, yielding tissue that more closely mirrors native muscle in morphology, gene expression, and contractile performance.

#### 2.2.1. Scaffold Selection and Microarchitectural Design

Engineering functional EMTs in vitro requires re-creating the native muscle’s aligned fibers within a supportive microenvironment. Key strategies include (1) using electrospun fiber scaffolds to direct myofiber orientation, (2) incorporating porous or hydrogel-based scaffolds to promote cell infiltration and ECM deposition, and (3) applying advanced bioprinting techniques combined with small molecule delivery systems to promote spatial organization and enhance tissue maturation.

Electrospinning has been used for fabricating ECM-mimicking fibrous scaffolds for decades. It controls fiber orientation and guides myoblasts to align and fuse into linear myotubes, similar to in vivo skeletal muscle fiber arrangement [42,43,44]. Studies have shown that C2C12 myoblasts seeded on aligned polymer nanofibers, like polycaprolactone (PCL) or polylactic acid (PLA), adhere and orient along the fibers within minutes and then differentiate into long, multinucleated myotubes parallel to the fiber axis [45,46,47]. However, traditional electrospun scaffolds often form dense fiber meshes that limit cell infiltration, only allowing cells to grow on the top layer of fibers [48]. This can yield tissue constructs that are thin and lack volumetric cell distribution, like other 2D patterned substrates. To achieve truly 3D muscle bundles, researchers have modified electrospinning techniques and scaffold designs. For example, increasing fiber spacing or pore size allows cells to migrate throughout the thickness of the scaffold. Phase-separation electrospinning can create fibers with internal pores, improving porosity without sacrificing alignment [49]. Another approach is layering aligned fiber mats or rolling them into tubes. Jana and Zhang directly electrospun aligned nanofibers into a tubular structure, creating a 3D aligned scaffold that supported circumferential cell alignment and myotube formation throughout the tube wall [50]. Such 3D fiber constructs overcome the diffusion and infiltration limits of flat mats, resulting in thicker skeletal muscle-like tissues.

Electrospun fiber scaffolds can be made from both synthetic polymers and natural polymers. As shown in Table 1, aligned fiber scaffolds made from diverse materials (synthetic polymers like PLGA/PCL or natural polymers like fibrin, chitosan, and alginate) consistently induce anisotropic myotube organization. Aligned scaffolds also often lead to improved twitch and tetanic contractile function in vitro compared to non-aligned constructs, due to the uniform orientation of force-generating myofibers [51,52]. Table 1 compares various electrospun fiber scaffolds, cell types, and key outcomes in EMT models.

While electrospun fibrous scaffolds offer directionally aligned platforms for differentiated skeletal muscle cells, porous hydrogels and sponge-like scaffolds provide a 3D environment that supports effective cell infiltration, nutrient diffusion, and new ECM deposition [53]. Hydrogels are high-water-content polymer networks that enable cells to establish 3D morphologies similar to in vivo conditions and enable efficient exchange of nutrients, oxygen, and signaling molecules throughout the construct [54].

**Table 1 cells-14-00882-t001:** Scaffold materials used for representative in vitro skeletal muscle research.

2D or 3D	Scaffold Materials	Cell Type/Model Used	Contractility Measurement	Other Key Findings	Advantages	**Disadvantages**	**References**
2D; Thin fibrous meshes (~0.1–0.3 mm thick)	Poly(lactide-co-glycolide) PLGA	C2C12	N/A	• Aligned PLGA fibers induced myoblast elongation, alignment, and differentiation into multinucleated myotubes. • Enhanced differentiation on aligned fibers (higher fusion index vs. random fibers/glass controls). • No biologic components (e.g., collagen/Matrigel) required for adhesion or differentiation	Synthetic, biodegradable scaffold with tunable degradation (PLGA); Compatible with long-term culture (no pH shifts or detachment issues)	No direct contractility measurement; Lack of cell infiltration due to dense fibrous structure	[43]
2D; Electrospun fibrous mats (quasi-3D environment)	Poly(butylene 1,4-cyclohexanedicarboxylate) (PBCE) and copolymers with triethylene cyclohexanedicarboxylate [P(BCE-co-TECE)]	C2C12	N/A	• The copolymer with higher “TECE” content (P73) promoted better C2C12 proliferation, alignment, and differentiation. • In vivo, the P73 scaffold became vascularized and integrated into both healthy and injured/dystrophic mouse muscle with minimal inflammation	Chemical tunability; Suitable for muscle repair	No direct contractility data	[52]
2D with 3D-like environment	Poly-IL-lactic acid (PLLA) nanofibers (~500 nm diameter; ~150 µm thick)	C2C12	N/A	Aligned nanofibers promote global myotube alignment, greater myotube length, and enhanced differentiation	Mimic native ECM and drive robust, aligned myotube formation.	Limited to thin-film culture; no direct functional (contractility) data.	[46]
2D with nanofibrous mat	Nanofibers made from a blend of polycaprolactone (PCL) and polyaniline (PANi)	C2C12	N/A	Aligned nanofibers and increased PANi content synergistically promote enhanced myotube alignment, longer myotubes, and upregulated myogenic genes	Mimics native ECM topography; incorporates electrical stimulation to boost differentiation; tunable via PANi concentration	Lacks direct contractile function	[47]
2D with thin nanofibrous scaffold (~10–15 μm thick)	PCL blended with 4% gelatin	C2C12	Assessed by calcium imaging (luminescence) under high-K^+^ and electrical stimulation (0.2–2 Hz). Mechanically stimulated aligned scaffolds showed enhanced contractility	Mechanical stimulation increases myotube width, actin density, and contractile function	Biomimetic nanofibrous architecture promotes adhesion and differentiation; mechanical stimulation further enhances regeneration	Low force production at high frequency stimulation in 2D	[55]
3D tubular construct	Chitosan/polycaprolactone (CS/PCL) blend	C2C12	N/A	Direct electrospinning of aligned 3D nanofibrous tubes; The inner surface supports formation of densely aligned myotubes mimicking native muscle architecture	3D fabrication with tunable scaffold dimensions	Limited function assessment; Long-term degradation	[50]
3D-bioprinted muscle tissue	GelMA-based bioink	C2C12	Under electrical stimulation (1 Hz), the engineered tissues generated a twitch contraction force of 443.085 μN for pennate muscle versus 239.662 μN for parallel muscle	• Pennate design (15° fiber orientation) significantly improved myotube alignment (51.93%) and enhanced contractile performance. • High cell viability (~79.89% by day 7) and uniform cell distribution were achieved.	3D bioprinting allows precise, customizable fabrication with complex microstructures; Direct contractility measurement	Use of murine C2C12 cells may not fully represent human muscle functionality.	[51]
3D engineered muscle	Fibrin-based gel	Primary rat myoblasts	Twitch force is 329 ± 26.3 μN and tetanic force is 805.8 ± 55 μN under electric stimulation	• Normal length-tension and force-frequency relationships; 50% force increase in response to IGF-I.	• Rapid, scaffold-free self-organization with minimal extracellular material. • Functional properties that mimic native muscle physiology. • Culture up to 6 weeks	Weaker contractile force due to the soft fibrin-based hydrogel	[56]
3D	Fibrin hydrogels enriched with Laminin-111	C2C12	N/A	• LM-111 enrichment produces a highly fibrous 3D architecture with thinner, more interconnected fibers • Lower Young’s modulus (2–6 kPa) favorable for myoblast proliferation • Enhanced VEGF, MyoD, and desmin expression, reduced IL-6 and myogenin at high LM-111 levels	Biomimetic ECM: mimics basal lamina composition and structure, and promotes a pro-regenerative secretory profile to supports cell adhesion, migration, and proliferation	High LM-111 (>500 µg/mL) impairs gel stability	[57]
3D	Fibrin + 10% Matrigel	Human primary myoblasts + HUVECs	N/A	Aligned myofibers and endothelial networks with 2 × 10^6^ total cells (50–70% muscle cells) Matrigel addition leads to uneven myofiber distribution.	Fibrin is proangiogenic and supports simultaneous myofiber alignment and vascular network formation	No direct contractility; HUVECs culture medium results in thinner myotubes.	[58]
3D	Collagen-1 + fibrin gel	Primary rat myoblasts	N/A	Hybrid gel prevents contraction; Higher cell proliferation in low-fibrin groups.	Hydrogel is biocompatible and mimicking natural ECM for myoblast differentiation.	Gel instability over time. Protease inhibitor, aprotinin is required.	[59]
3D	Collagen sponges/OPLA (open-cell poly-lactic-acid) scaffolds	Primary rat myoblasts	N/A	High apoptosis (45–66% in sponges); OPLA lacks elasticity for muscle.	Sponges is high stability and synthetic OPLA is biodegradable.	Poor cell viability. • Unsuitable pore structure (sponges). • Rigid (OPLA).	[59]
3D	Decellularized skeletal muscle ECM (mdECM) bioink + PCL constraints	C2C12	Observed spontaneous contractions upon eletrical stimulation without quantitative measurement	Elastic modulus ≈ 12 kPa (similar to native muscle); Striated myotubes formed.	High cell viability (>90%); Customizable architecture via 3D printing.	Requires specialized 3D printing setup.	[60]

Many natural hydrogels (e.g., collagen, fibrin, and gelatin) supply integrin-binding ligands and a bioactive milieu that myogenic cells can remodel and enrich with de novo ECM; by contrast, polysaccharide-based gels such as alginate, agarose, or unmodified hyaluronic acid are intrinsically non-adhesive and must be chemically modified (commonly with RGD peptides or other motifs) to support skeletal muscle cell attachment and matrix deposition (Table 1). For instance, fibrin has been extensively used as a scaffold in engineered tissues. A previous study revealed that smooth muscle cells embedded in fibrin gels can degrade the fibrin and replace it with their own collagenous matrix within 2–4 weeks [61]. Fibrin also contains multiple heparin-binding domains, allowing it to sequester and slowly release growth factors important for skeletal muscle regeneration [56,57]. Collagen-based hydrogels are extensively utilized due to their compositional and structural similarity of native ECM components [59,62] (Table 1). However, collagen-I hydrogels exhibit significant contraction (up to 80% area reduction) under mechanical tension due to cell-mediated remodeling [63]. Alternatively, gelatin, a denatured collagen derivative, retains RGD motifs that are critical for cell adhesion but has reduced contraction risk, enabling scalable fabrication of bioactive scaffolds [55].

Decellularized ECM (dECM) scaffolds derived from native tissues (such as skeletal muscle, small intestine submucosa (SIS), or bladder matrix) offer a protein-rich, aligned architecture that is difficult to replicate synthetically. When the decellularization process preserves collagen alignment and basement-membrane proteins, these matrices can serve not only as implants but also as 3D substrates for myotube culture in vitro [60,64]. For instance, Perniconi et al. seeded C2C12 myoblasts onto a whole-organ muscle acellular scaffold [65]. The cells infiltrated the porous matrix, aligned along native collagen bundles, and fused into contractile, multinucleated myotubes within one week of culture. dECM biomaterials are highly bioactive and immunomodulatory as they break down in vitro. They release chemoattractants, cytokines, and extracellular vesicles that recruit endogenous progenitor cells and polarize immune cells towards a regenerative phenotype [66,67,68]. Clinical studies with dECM scaffolds for skeletal muscle repair have demonstrated promising outcomes. Sicari et al. implanted an injectable SIS-derived hydrogel into patients with volumetric muscle loss and three out of five patients showed marked functional improvements in the treated area [69]. In a larger 13-patient study, implantation of an ECM scaffold led to an average ~37% increase in strength of the injured muscle over 6 months, along with increased exercise capacity [70]. These results highlight the translational potential of dECM hydrogels, which can accommodate irregular wound geometries and encourage the patient’s own cells to rebuild skeletal muscle tissue.

Synthetic hydrogels (e.g., PEG, polyacrylamide, pluronic, etc.) have emerged as a central focus in the development of biocompatible and tunable scaffolds for volumetric muscle loss (VML) repair. They provide precise control over mechanics and degradation but lack intrinsic bioactivity [71]. For skeletal muscle engineering, synthetic gels are typically modified with cell-adhesion peptides (e.g., RGD peptides) or combined with natural components to support myogenic cell attachment [72]. Studies demonstrate that myoblasts cultured on PEG substrates with a stiffness of ~12 kPa exhibit enhanced actin/myosin striations and differentiation compared to softer or stiffer substrates [73]. Furthermore, photo-polymerizable PEG diacrylate (PEGDA) hydrogels enable spatial patterning of biochemical molecules, such as growth factor or ECM proteins, to study localized effects on myogenic alignment and fusion [74]. In 3D contexts, previous studies using tunable synthetic or bioprinting hydrogels have emerged as a promising platform for reconstructing functional muscle tissue. When implanting into murine VML models, these constructs facilitated significant functional recovery by promoting vascularization, innervation, and de novo myofiber formation within the defect site [75,76]. Notably, a study by Kim et al. demonstrated that the 3D-printed PCL scaffold seeded with human primary muscle progenitor cells restored 82% muscle force in rat VML model at 8 weeks of post-implantation [77].

#### 2.2.2. Perfusion System Integration

A general challenge for thick, engineered tissues is the transportation of oxygen and nutrients during culture. Diffusion alone is sufficient only for tissues up to ~100–200 μm from a nutrient source [78,79]. Beyond this limit, cells become hypoxic and necrotic unless a perfusion system is introduced [79]. EMTs thus often require perfusion systems or vasculature analogues to sustain cell viability throughout the constructs.

Perfusion bioreactors have been used to overcome diffusion limits by actively flowing culture medium through engineered tissues. In one of the earliest studies, Chromiak et al. cultured 3D skeletal muscle organoids in a perfusion bioreactor and showed stable metabolic activity for up to 30 days by measuring metabolized medium glucose and skeletal muscle tissue protein synthesis and degradation rates [80]. Total protein/DNA content decreased 22–28% over a 13-day period [80].

Another perfusion approach involves using hollow-fiber bioreactors by circulating media through semi-permeable capillary-like fibers within the construct to supply nutrient and oxygen. Yamamoto et al. designed a hollow-fiber perfusion system to culture C2C12 myoblasts within a 3D type I collagen matrix, achieving high-cell-density skeletal muscle tissues [81]. The results show a dense, viable skeletal muscle construct containing multinucleated myotubes, as confirmed by histology [81].

Moreover, microfluidic organ-on-chip platforms have been developed to cultivate small skeletal muscle tissues under continuous perfusion in microscale chambers. Agrawal et al. established a skeletal muscle-on-a-chip model that recapitulates muscle architecture and can be continually perfused with culture medium [82]. In their design, C2C12 myoblasts embedded in a gelatin hydrogel self-aligned between two flexible micropillars inside a microfluidic device. The chip is constantly perfused during culture. As the cells differentiated, they formed functional myofiber bundles that generated measurable passive tension between the pillars. Importantly, this perfusion model was used for studying injury by treatment with a muscle toxin. They observed that higher doses of this muscle toxin cause tissue damage leading to changes in the EMT’s structure and function [82].

#### 2.2.3. In Vitro Neuromuscular Junction (NMJ) Formation and Integration into Engineered Models

One hallmark of mature skeletal muscle is its innervation by motor neurons, forming NMJs [83]. In EMTs, promoting NMJ formation is key to achieving physiological functions and muscle maturation. Co-culturing skeletal muscle cells with motor neurons or adding exogenous factors can drive the assembly of NMJs in vitro. Bakooshli et al. described a 3D model where human skeletal muscle progenitors were mixed with pluripotent stem cell-derived motor neurons (MNs) in a fibrin gel [18]. Over a few weeks, the MN axons extended and formed functional NMJ connections with the skeletal muscle fibers. Calcium transients were measured to confirm the integration of the NMJ. In this study, a 3D co-culture system enhanced fiber maturation by showing significantly increased myotube diameters and diverse fiber types and higher acetylcholine receptors (AChR) when compared with a 2D co-culture system [18]. Another group demonstrated a human 3D neuromuscular model for amyotrophic lateral sclerosis (ALS) research, where myoblasts and iPSC-derived MNs were co-cultured in a silicone dish with hook-and-loop fastener anchors to support skeletal muscle bundle formation under tension [84]. This system achieved robust myofiber formation including AChR clustering, detected by α-bungarotoxin staining and visible neurite innervation at endplates [84].

Integration of NMJs into bioengineered muscle also opens doors for disease modeling (e.g., myasthenia gravis or motor neuron diseases) and drug testing on a “motor unit” level. A recent study of neurospheres co-cultured with a 3D EMT model demonstrated loss of muscle function in a dose- and time-dependent manner after botulinum neurotoxin treatment [40]. Such models are invaluable for studying synaptic defects and screening therapies that aim to restore neuromuscular function.

#### 2.2.4. Co-Culture Strategies with Supporting Cell Types

Native skeletal muscle comprises a heterogeneous tissue architecture consisting of multinucleated myofibers and diverse non-muscle cell populations, including fibroblasts, endothelial cells, pericytes, neurons, and macrophages, which collectively orchestrate critical processes such as myogenesis, tissue repair, and metabolic homeostasis [85,86].

Co-culture strategies in 3D models aim to incorporate these supporting cell types to create a more biomimetic and functional tissue. Fibroblasts are crucial for ECM production and maturation of skeletal muscle fibers. In EMTs, adding a small fraction of fibroblasts (often 10–20% of total cells) can greatly improve EMT contractile function [16,87]. Human primary fibroblasts included with myoblasts in a fibrin matrix helped produce thicker, more robust myofibers and prevented tissue rupture during contraction [16,38]. Fibroblasts deposited additional collagen and fibronectin, contributing to a more mature ECM that supports force transmission.

Endothelial cells are another popular cell type to add to EMTs, given muscle’s rich capillary network. Gholobova et al. established that co-culturing human primary myoblasts with endothelial cells facilitates the spontaneous formation of microvascular networks within EMTs [58]. Human myoblasts were combined with human umbilical vein endothelial cells (HUVECs) at a 1:1 ratio within a fibrin hydrogel matrix, which was subjected to mechanical tension to promote alignment of skeletal muscle fibers. Following optimization of culture parameters, the resulting constructs exhibited densely packed, well-aligned myofibers interspersed with endothelial cell-derived tubules, forming interconnected vascular-like networks. The presence of endothelial networks improved muscle viability and may have induced some angiogenic growth factors from muscle cells themselves. Co-cultured endothelial cells and myoblasts engage in bi-directional signaling, with endothelial cells secreting VEGF and HGF growth factors that stimulate myogenesis, while differentiated myotubes release paracrine signals including VEGF and angiopoietin-1, which stabilize nascent vascular networks and enhance endothelial cell survival [88,89]. This crosstalk recapitulates the in vivo interdependence of myogenesis and angiogenesis that is critical for functional skeletal muscle regeneration. Tri-culture of C2C12, HUVECs, and fibroblasts in 3D scaffolds promote both striated myotubes and vascular network formation after 3 weeks of perfusion in in vitro culture. When these pre-vascularized EMT constructs were implanted in vivo to replace full-thickness abdominal wall defects in nude mice, blood perfusion was significantly improved compared to constructs without endothelial cells [90].

Other supporting cells used in co-culture include pericytes or mesenchymal stromal cells (MSCs), which can differentiate into fibro-adipogenic cells or provide trophic support to muscle progenitors. MSCs enhance the formation of vascular networks when combined with endothelial cells in scaffolds [91,92]. In EMTs, MSCs or adipogenic cells can model pathological states like fatty infiltration. Shahin-Shamsabadi et al. engineered a self-assembled 3D collagen model in which precisely positioned 3T3-L1 preadipocytes adjacent to C2C12 myoblasts were either in direct physical contact or separated by a ~300 μm collagen spacer to study obesity-related skeletal muscle changes [93]. In the direct-contact co-culture, skeletal muscle cells exhibited atrophy related to altered protein expression, and adipocytes themselves accumulated larger lipid droplets mimicking intramuscular lipid-like features. In the indirect co-culture set-up, adipocytes exhibited higher basal and drug-induced glycerol release, indicating enhanced lipolysis, while the proteostasis phenotype was attenuated compared to direct co-culture setting [93]. These results show that not only the presence of adipocytes but also their spatial arrangement relative to skeletal muscle fibers dictates whether lipid is stored or mobilized in a skeletal muscle co-culture system.

### 2.3. Modeling Muscular Disease in 3D Constructs

#### 2.3.1. Mendelian Muscular Disorders

Advances in 3D muscle engineering have enabled modeling of inherited myopathies, which are muscular diseases caused by germline mutations. Traditional 2D cell models often fail to capture key pathological features, especially tissue-level contractile functional deficits, but 3D EMTs can recapitulate many aspects of disease physiology.

Duchenne muscular dystrophy (DMD), one of the most prevalent and severe X-linked neuromuscular disorders, is characterized by progressive skeletal muscle degeneration, cardiomyopathy, and respiratory failure due to mutations in the *DMD* gene, which encodes the cytoskeletal protein dystrophin [94]. A hallmark of DMD pathology is instability, where dystrophin deficiency destabilizes the skeletal muscle membrane, rendering fibers susceptible to contraction-induced damage and necrosis [95]. Recent advances in 3D in vitro models have enabled recapitulation of these pathological mechanisms to accelerate preclinical research.

Tejedera-Villafranca et al. generated a 3D skeletal muscle model by embedding DMD patient-derived immortalized myoblasts within fibrin hydrogel [96]. Their constructs initially exhibited normal myotube differentiation but developed membrane ruptures and viability loss under 30 min 1 Hz electrical pulse stimulation, mirroring in vivo contraction-induced injury. Functionally, DMD bundles displayed progressive fatigue with reduced tetanic force retention. Treatment with a utrophin-upregulating compound partially restored membrane integrity, highlighting the model’s utility for screening dystrophin-independent therapies [97].

Complementing this, Ebrahimi et al. established immortalized myoblast lines derived from both healthy donors and individuals with DMD and encapsulated them within a 3D fibrin–Matrigel hydrogel system [98]. Their model revealed impaired calcium handling in DMD constructs and identified spontaneous “revertant” myofibers, which are rare skeletal muscle fibers that naturally restored dystrophin expression. This partial functional recovery mirrors observations in human DMD patients as previously documented [99], and this study represents the first in vitro recapitulation of this phenomenon.

Moreover, another recent study demonstrated a rapid approach for generating an isogenic DMD model by knocking down dystrophin via shRNA in a healthy hiPSC-derived myogenic progenitor cell line. The dystrophin-deficient 3D EMTs showed loss of contractile force and disorganized tissue architecture [100]. Impressively, delivering a micro-dystrophin gene to these tissues partially restored their function, validating the model’s responsiveness to a known therapeutic strategy.

Limb–Girdle Muscular Dystrophies (LGMDs) are a heterogeneous group of genetic myopathies affecting hip and shoulder muscles, often due to mutations in sarcolemmal or extracellular matrix proteins [101]. Limb–Girdle Muscular Dystrophy Type 2A (LGMD2A), caused by mutations in the *CAPN3* gene encoding calpain-3, has recently been modeled in vitro in EMT [100]. Groen et al. established a rapid and versatile model for LGMD2A by integrating a lentiviral short hairpin RNA (shRNA) strategy into human iPSC-derived myogenic progenitors. Notably, these *CAPN3*-deficient EMTs exhibited hallmark pathological features of LGMD2A, including disrupted myofiber organization, progressive loss of structural integrity, and severely diminished contractile function. Proteomic analysis of these constructs further revealed disease-specific molecular signatures characterized by downregulation of proteins involved in skeletal muscle contraction, proteasomal function, and cytoskeletal integrity, mirroring findings from LGMD2A patient biopsies and animal models.

Another recent study about LGMD in 3D culture was focused on modeling the phenotype of LGMD2B, a severe dysferlinopathy characterized by progressive skeletal muscle weakness. Khodabukus et al. developed an EMT using human hiPSC-derived skeletal muscle cells to investigate the pathogenesis and underlying mechanisms of LGMD2B [102]. Utilizing hiPSC-derived myogenic progenitor cells harboring dysferlin mutations, they demonstrated that engineered LGMD2B EMTs exhibit impaired contractile function and defective calcium handling compared to healthy controls. Specifically, these LGMD2B EMTs displayed reduced calcium transient amplitudes during tetanic contractions and significant deficits in membrane repair capabilities following induced hypo-osmotic injury, confirming critical aspects of dysferlin deficiency observed clinically [102]. Transcriptomic analysis further revealed that LGMD2B EMTs had substantial downregulation of genes associated with oxidative metabolism, mitochondrial respiration, and intracellular calcium regulation, highlighting mitochondrial dysfunction as a significant contributor to disease pathology. They further identified a pathogenic role of cytosolic calcium leaking through excessive ryanodine receptor (RyR1) activity, and pharmacological interventions with the RyR inhibitor dantrolene or the dissociative glucocorticoid vamorolone effectively restored LGMD2B EMT contractility, improved membrane repair, normalized mitochondrial function, and reduced lipid accumulation [102].

Myotonic Dystrophy type 1 (DM1) is an autosomal dominant disorder caused by a CTG repeat expansion in *DMPK* [103]. It primarily affects skeletal muscle strength and mass by causing RNA splicing defects. Fernandez-Garibay et al. established a bioengineered 3D model of DM1 using transdifferentiated myoblasts from DM1 patient-derived fibroblasts [104]. The DM1 EMTs reproduced hallmark features, the formation of CUGexp-RNA foci, and co-localization with alternative splicing regulator MBNL1. Such features are difficult to observe in 2D culture. They also observed smaller myotube diameters in DM1 constructs compared to healthy controls, reflecting the atrophy seen in patients. Furthermore, when these diseased constructs were treated with antagomiR-23b [105], the molecular defects were rescued as well as increased myotube size [104].

Facioscapulohumeral Dystrophy (FSHD) is caused by aberrant expression of the *DUX4* gene which is normally silenced in skeletal muscle [106]. A recent work used human iPSCs from mosaic FSHD1 patients in a 3D construct to investigate *DUX4*-mediated pathology [107]. Their model recapitulates key disease features, including sporadic *DUX4* expression, reduced absolute contractile forces, and smaller myofiber diameters, while maintaining sarcomeric structure and differentiation markers. When compounds that suppress *DUX4* in 2D cultures (pamapimod, CK1 inhibitor, and rebastinib) were screened in EMTs, they showed minimal DUX4 inhibition and failed to rescue contractility.

Pompe Disease is an inherited metabolic myopathy characterized by abnormal glycogen storage due to lack of the enzyme acid alpha-glucosidase (GAA) in skeletal muscle lysosomes [108]. Conventional in vitro models, while replicating intracellular glycogen storage, fail to mimic functional aspects of muscle physiology such as contractility and fatigue dynamics. A recent study using 3D skeletal muscle models derived from patient-specific iPSCs has demonstrated that glycogen accumulation disrupts mechanical properties, including increased tissue stiffness and accelerated fatigue during electrical stimulation, likely due to impaired energy metabolism [109]. Notably, therapeutic intervention with recombinant human GAA (rhGAA) in these 3D systems effectively reduces glycogen deposits and restores contractile function, mirroring clinical improvements observed in enzyme replacement therapy-treated patients.

Emery–Dreifuss muscular dystrophy (EDMD) is a rare inherited myopathy characterized by early-onset contractures of major tendons, progressive skeletal muscle wasting, and cardiomyopathy, most often caused by mutations in the nuclear envelope proteins emerin (*EMD*) or lamin A/C (*LMNA*) [110]. To overcome the limitations of 2D cultures in modeling such complex phenotypes, several groups have developed 3D EMTs that more faithfully recapitulate EDMD pathology [111,112]. Patient-derived induced pluripotent stem cells (iPSCs) carrying *LMNA* mutations differentiated into 3D fibrin hydrogel-based EMT exhibit hallmark pathological features, including a 2–3-fold increase in nuclear envelope abnormalities (e.g., elongated or fragmented nuclei) compared to isogenic controls. These constructs also demonstrate functional deficits, generating 25–40% lower contractile force under electrical stimulation, mirroring the progressive muscle weakness observed in patients [111]. Extending this approach to cardiac tissue, engineered heart microtissues from EDMD iPSC-cardiomyocytes display arrhythmic contractions and diminished twitch stress linked to suppressed JAK2/STAT3 signaling, and pharmacological activation of STAT3 partially restores contractile function [113].

In general, monogenic muscle disorders are increasingly being modeled using EMTs. Table 2 describes their creation and how well they recapitulate disease features, as well as any therapeutic interventions tested. The key advantage of 3D models is the ability to observe tissue-level properties of the disease that are not apparent in 2D. This makes them powerful platforms for mechanistic studies and preclinical drug testing.

#### 2.3.2. Polygenic (Common) Muscle Diseases

Musculoskeletal pathologies frequently arise from complex, multifactorial pathological mechanisms rather than monogenic origins. Three-dimensional skeletal muscle models are now being applied to such polygenic or acquired conditions, like age-related sarcopenia and obesity-related skeletal muscle degeneration. While these systems face limitations in fully recapitulating human disease complexity, these models permit the controlled study of individual factors on muscle function.

Sarcopenia is the progressive loss of skeletal muscle mass and strength with age. It involves hormonal changes, chronic inflammation, and altered muscle stem cell function [120]. Giza et al. developed a microphysiological 3D muscle-on-chip using human myoblasts from young vs. old donors [114]. After 5 days of 30 min low-frequency electrical stimulation that mimicked “daily physical activity”, young-adult derived EMTs showed a significantly higher magnitude of contractile force while constructs from sedentary older donors exhibited lower and non-uniform force production waveforms in response to electrical pacing [114]. Notably, daily “exercise” triggered a hypertrophic signaling pathway in the young adult tissues, as evidenced by a significant upregulation of IGF-1 and contractile-protein genes (*ACTN3* and *TNNI2*), whereas the old adult muscle chip showed little or no transcriptional response [114].

Similarly, in a recent study by Wang et al., a bioengineered 3D skeletal muscle model using mouse primary myoblasts was developed to investigate cell-autonomous mechanisms of age-related decline in regenerative capacity [115]. Adult mouse primary myoblasts were isolated from young and aged mice. EMTs with old mice cells exhibited hallmark features of aging, including myotube atrophy, reduced contractile force, and impaired regeneration following cardiotoxin-induced injury [115]. Transcriptomic profiling revealed an age-dependent upregulation of complement component 4b (C4b) in old EMTs, and pharmacological inhibition of C4b via complement factor I (CFI) administration enhanced the regenerative capacity after cardiotoxin (CTX) injury, including restoring myotube formation and contractile performance [115].

Aging is characterized by a chronic, low-grade systemic inflammation that is often called “inflammaging” in which circulating levels of pro-inflammatory cytokines and acute-phase mediators remain persistently elevated [121,122]. Engineered muscle-on-chips exposed to inflammatory stimuli demonstrate how anti-inflammatory interventions restore function. Chen et al. used a 3D human EMT model to study interferon-γ (IFN-γ)–induced skeletal muscle wasting [116]. IFN-γ exposure caused significant myofiber atrophy and weakness in the muscle constructs derived from multiple donors. However, electrical stimulation training prevents IFN-γ-induced structural and functional deterioration of EMTs. Additionally, a JAK inhibitor that blocks IFN-γ downstream signaling fully prevented the induced myopathy, confirming that blocking inflammatory pathways can rescue skeletal muscle tissue function in vitro [116].

Obesity and metabolic syndrome negatively impact skeletal muscle, causing insulin resistance, fat infiltration, and chronic inflammation in skeletal muscle tissue [123]. To investigate these changes, researchers have designed 3D co-culture models that incorporate skeletal muscle cells with adipocytes and metabolic stress models. Shahin-Shamsabadi et al. established a 3D co-culture of adipocytes and myoblasts to study direct versus paracrine interactions between fat and skeletal muscle [93].

To model lipid infiltration into skeletal muscle (a pathological phenomenon termed myosteatosis associated with obesity, diabetes, and aging), Hernandez et al. engineered a vascularized muscle tissue with embedded adipocytes to mimic intramuscular adipose tissue accumulation [117]. They started with primary skeletal muscle precursor cells and microvascular fragments from adipose tissue to form 3D EMTs that have their own capillary-like networks. By applying adipogenic differentiation media after an initial skeletal muscle maturation period, a portion of the supporting cells in the construct differentiated into adipocytes, creating a mixed muscle–fat tissue. Three-dimensional co-culture results show reductions in skeletal muscle fiber maturation and reduced tissue stiffness/elastic modulus with adipocytes. Metabolic measurements showed an elevated basal glucose uptake in adipocyte-rich skeletal muscle but a blunted insulin-stimulated glucose uptake, indicating insulin resistance when intramuscular fat is present [117]. This mirrors the situation in obese insulin-resistant skeletal muscle, where excess fatty deposits can cause chronic insulin signaling impairments.

Another approach to introduce fat accumulation is to treat EMTs with excess nutrients or free fatty acids (FFAs) to induce lipotoxic stress. Palmitate, a saturated fatty acid, is widely used to create insulin resistance in vitro. Na et al. recently reported a 3D skeletal muscle organoid-based screening system for type 2 diabetes drugs [118]. They embedded C2C12 myoblasts in a 3D alginate/collagen scaffold to form muscle stands and observed induced insulin resistance by exposing them to 0.1 mM palmitate. Compared to 2D cultures, EMT had a threefold higher glucose uptake response to insulin, and it maintained this sensitivity over longer-term differentiation, making the insulin resistance effect easier to detect. The team validated the platform by testing known anti-diabetic drugs: Rosiglitazone (a PPARγ agonist) and metformin both improved glucose uptake in the palmitate-treated EMTs, consistent with their expected actions [118]. Their findings highlighted that the EMT context provides more predictive results for metabolism-modulating drugs. However, the absence of contractility assessments in these studies restricted the ability to evaluate functional outcomes. To address this gap, a complementary investigation utilized C2C12-derived 3D EMT exposed to a FFA mixture (oleic, palmitic, linoleic, and α-linoleic acids [OPLAs]) at physiological (200 µM) and pathological (800 µM) concentrations after differentiation [119]. OPLA exposure triggered dose-dependent lipid droplet accumulation, upregulated lipid storage-associated genes (e.g., perilipin 2), and altered metabolic pathways. Critically, this study directly assessed contractile function, revealing a significant reduction in both twitch and tetanic forces following FFA treatment.

Collectively, these skeletal muscle disease modeling applications signify that 3D muscle disease models are bridging the gap between bench and bedside. They function both as test platforms for interventions and as discovery tools for understanding skeletal muscle pathophysiology. Table 2 summarizes disease modeling in EMTs, including the methods to create each model, the disease phenotypes observed, and any treatments applied to test for corrections of those phenotypes.

### 2.4. Current Challenges and Future Directions

#### 2.4.1. Technical Challenges

EMT models have historically relied on the murine C2C12 myoblast cell line due to its robust myogenic capacity and ease of use. However, C2C12-derived tissues have limited translational relevance, prompting a shift toward human cell sources [124]. Recent studies emphasize using human primary myoblasts or induced pluripotent stem cell (iPSC)-derived myogenic progenitors to create more clinically predictive skeletal muscle constructs [38,124]. Achieving in vitro skeletal muscle maturity remains challenging, as engineered fibers typically resemble embryonic or neonatal muscle in genotype and function. Extended culture time, biochemical molecule treatment, and electrical or mechanical stimulation are being explored to enhance myotube maturation [124,125,126]. Additionally, contractility measurements have become essential benchmarks of tissue quality, especially for disease modeling. Typically, EMTs are evaluated for twitch force (single stimulus contraction), tetanic force (maximal sustained contraction under high-frequency stimulation), and fatigue resistance (force decline under repeated stimulation) as key functional outputs [127]. Several platforms now enable these measurements, from sensitive force transducers to imaging of micropillar deflection, in both rodent and human skeletal muscle constructs (Table 2).

Another technical hurdle is the absence of key muscle-resident cell types in many 3D muscle models. Most EMTs to date consist predominantly of myogenic cells and fibroblasts for structural support [124]. In native skeletal muscle tissue, a variety of support cells—including fibro-adipogenic progenitors (FAPs), pericytes/endothelial cells, and immune cells—interact with myofibers to maintain homeostasis and aid regeneration. The lack of these cell types in vitro can limit the extent to which EMTs can accurately model in vivo tissue. Recent efforts highlight that incorporating non-myogenic cell populations can improve tissue assembly and function. For instance, co-culturing skeletal muscle progenitors with fibroblasts or FAP-like cells can enhance extracellular matrix deposition and skeletal muscle fiber alignment. Inclusion of endothelial cells and pericytes can promote pre-vascular network formation, addressing the critical issue of vascularization in larger constructs. Moreover, innervation is vital for skeletal muscle function and maturation. Despite these advances, achieving fully mature, vascularized, and innervated 3D human skeletal muscle tissue in vitro remains a grand challenge. First, each lineage, myogenic, endothelial, mesenchymal, and immune, requires distinct biochemical and mechanical conditions: myotube differentiation favors stiffer (~10–15 kPa) matrices and low serum media, whereas endothelial tubulogenesis and FAP survival often depend on softer hydrogels and VEGF-rich medium [128]. Designing a biomaterial scaffold capable of fulfilling these conflicting biological demands while avoiding detrimental crosstalk between co-cultured cell populations remains an unresolved challenge in muscle tissue engineering. In particular, the inclusion of immune cells and FAPs in 3D EMTs complicates system dynamics. These populations actively remodel ECM architecture and exhibit mechanoresponsive behaviors, with disruptions to homeostatic culture conditions commonly inducing fibrosis or ectopic adipogenesis. Ongoing work suggests that a combination of prolonged differentiation, multi-cell type co-culture, and biomimetic stimuli will be required to reach the structural and functional maturity seen in vivo.

#### 2.4.2. Scalability

EMTs are typically millimeters in length/diameter, a scale limited by both fabrication methods and nutrient diffusion constraints. Scaling up to centimeter-sized or volumetric muscle constructs presents significant difficulties. A primary issue is sustaining cell viability and function throughout large constructs. Because oxygen in dense tissues such as skeletal muscle diffuses only ~100–200 µm from its source, cells located beyond that distance become hypoxic within hours, and, if diffusion is not restored, they will undergo necrosis [129]. To overcome this limitation in oxygen diffusion, studies have explored perfusion bioreactors and pre-vascularization strategies. For example, Borisov et al. developed a perfused 3D EMT 3 mm thick and 8 mm in diameter and further upscaled it to a 50 mm diameter construct by seeding with stromal vascular fraction cells to support angiogenesis [129]. Future directions to improve scalability by size include bioprinting channels for perfusion, using oxygen-releasing biomaterials, and developing composite tissues that integrate skeletal muscle with vascular beds.

#### 2.4.3. Future Directions

Three-dimensional bioprinting is an emerging approach to fabricate skeletal muscle tissues with spatial precision. Bioprinting allows the placement of cells and bioinks in predefined architectures. For instance, it enables the printing of parallel micron-scale filaments of skeletal muscle progenitors encapsulated in hydrogel to mimic the alignment of skeletal muscle fibers [125]. Recent advances in bioprinting of skeletal muscle have demonstrated viable muscle constructs with perfusable vascular channels and even integrated nerve guide scaffolds in the same print, aiming to pre-organize the tissue for faster vascularization and innervation post-implantation. Concurrently, organ-on-chip technology is being applied to skeletal muscle tissues. Microfluidic platforms can house small muscle bundles and perfuse them with media or for drugs screening, closely controlling the microenvironment. These chips often include stretch mechanisms to simulate exercise and can connect skeletal muscle tissue with other organ tissues. For example, multi-organ platforms have coupled skeletal muscle on a chip with intestine, liver, kidney proximal tubule, and blood–brain barrier equivalents in vitro [130,131,132]. Such systems enable investigations into organ crosstalk under near-physiological conditions.

In future work, it will be essential to integrate molecular-level monitoring tools into both 2D and 3D skeletal muscle culture platforms. Complementary endpoint analyses, such as spatially resolved proteomics and single-cell transcriptomics, will offer comprehensive insights into pathway activation and differentiation trajectories at discrete developmental stages. For example, Reggio et al. [133] recently illustrated that a PEG–fibrinogen 3D scaffold creates a low-proliferation niche by downregulating mitogenic cascades (RAS–RAF–MAPK and EGFR) while upregulating myogenic transcription factors (MYOD1 and MYOG) and sarcomeric proteins (myosin and troponin) as revealed by mass spectrometry. Such proteomic shifts emphasize that molecular responses in 3D deviate markedly from those in standard monolayers, highlighting the need for both live monitoring and comprehensive endpoint profiling to guide scaffold design and culture protocols. By combining noninvasive, continuous readouts with detailed omics characterization, researchers can systematically refine hydrogel composition and dynamic culture conditions.

Looking forward, the convergence of these emerging technologies could dramatically enhance EMTs by promoting greater structural maturation, enhancing vascularization and innervation. Bioprinting and microfluidics may allow for the construction of large, complex skeletal muscle organs complete with intrinsic vasculature and nerves, while iPSC and gene editing technologies provide an unlimited supply of patient-specific skeletal muscle cells and the means to modify them. As these tools develop, EMTs are expected to become ever more biomimetic, recapitulating not only the structure and function of native skeletal muscle, but also its regeneration capacity and fidelity to human diseases.

## Figures and Tables

**Figure 1 cells-14-00882-f001:**
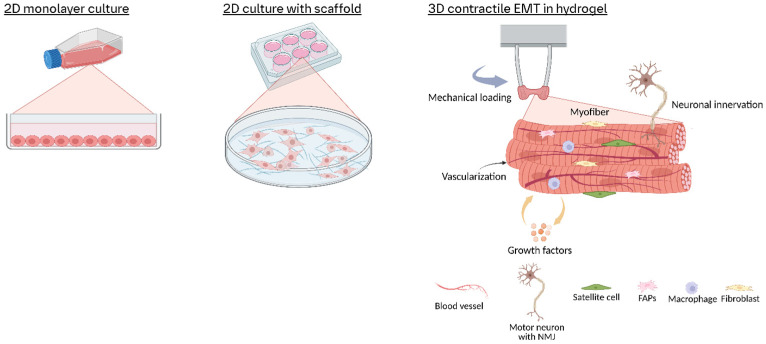
Schematic overview of in vitro skeletal muscle culture models. In vitro skeletal muscle systems range from 2D monolayers or scaffold-guided cultures of aligned myotubes to 3D hydrogel constructs cast between posts. These 3D tissues develop tension, enable force measurement, and support co-cultures with progenitor, stromal, immune, and neural cells to recapitulate native muscle architecture and function.

**Table 2 cells-14-00882-t002:** Muscular diseases modeled in 3D engineered muscle tissue.

Disease Model	Cell Types	Approach of 3D Muscle	Recapitulated Disease Features	Therapeutic Interventions Tested	**Limitations**	**References**
Duchenne Muscular Dystrophy (DMD)	Immortalized human muscle precursor cells (from both healthy controls and DMD patients)	Fibrin–Matrigel composite hydrogel cast in PDMS molds with flexible posts	The 3D model mimics key DMD features: sarcolemmal damage and reduced tetanic contraction and less resistance to fatigue.	Tested utrophin up-regulators (Ezutromid, Halofuginone, and a novel SOMutrophin candidate); only SOMutrophin increased utrophin nearly three-fold and improved contractile forces	High inter-individual variability, incomplete recapitulation of all DMD phenotypes (especially sarcolemmal stability).	[96]
	Immortalized human myoblasts (isolated from DMD patient biopsies and from healthy donors)	Fibrin-based hydrogel with Geltrex, cast in the MyoTACTIC micro-mold device	• Absence of dystrophin in most DMD fibers. • Rare “revertant” dystrophin-positive fibers in one DMD line • Altered calcium handling vs. healthy lines	β1-integrin activating antibody (TS2/16) • Improved sarcomere organization • Some protection from contraction-induced structural damage	3D muscles were not fully mature, and culture time was relatively short.	[98]
	Human iPSC-derived myogenic progenitor cells	Fibrin + 20% Matrigel composite hydrogel cast in PDMS molds with flexible posts	• DMD knockdown was generated via shRNA-mediated knock down delivered via lentiviral transduction • Near-zero twitch force in DMD KD at day 9	Co-express micro-dystrophin via lentiviral transduction. Partial rescue of contracile force in DMD KD 3D muscle.	Not complete knock down or restore of DMD; Lack of other resident cell types in native muscle.	[100]
Limb-Girdle Muscular Dystrophies type 2A (LGMD2A)	Human iPSC-derived myogenic progenitor cells	Fibrin + 20% Matrigel composite hydrogel cast in PDMS molds with flexible posts	• Lentiviral shRNA knockdown targeting *CAPN3* in hiPSC-derived MPCs before 3D tissue formation. • Severe reduction in contractile force, disrupted tissue architecture, and proteomic changes consistent with CAPN3 deficiency	N/A	Knockdown rather than complete KO; short culture period.	[100]
LGMD2B	hiPSC-derived myogenic progenitor cells	Fibrin + Matrigel hydrogel	• Severely reduced force generation (50%) Diminished Ca^2+^ transient amplitude • Deficient membrane repair; • Impaired mitochondrial function under fatty acid treatment and lipid droplet accumulation.	RyR Ca^2+^ channel inhibitor (dantrolene), novel glucocorticoid (vamorolone); Improved structure and contractile functions	Disease severity in iPSC model can differ from native tissues; does not model all late-stage or systemic features of LGMD2B.	[102]
Myotonic Dystrophy Type 1 (DM1)	Immortalized transdifferentiated human myoblast-like cells from DM1 fibroblasts	Gelatin methacryloyl + carboxymethyl cellulose methacrylate (GelMA-CMCMA)	• No direct contracile force • 3D micropatterning improves myogenic fusion. • Thinner myotubes in DM1	Antisense oligonucleotide (antagomiR-23b)	No functional contractility readout	[102,103]
Facioscapulohumeral Dystrophy (FSMD)	hiPSC-derived myogenic progenitors from mosaic FSHD1 patients	Fibrin + Matrigel hydrogel in custom PDMS “T-bone” molds	FSHD3D muscle: upregulated *DUX4* & target genes; Reduced absolute force with myofiber atrophy and smaller sarcomeres.	p38 inhibitor (pamapimod), CK1 inhibitor, & rebastinib tested daily; showed no function and fiber size improvement in 3D	No co-cultured immune or supporting cells; immature 3D muscle	[107]
Pompe disease	Primary myoblasts from infantile-onset Pompe (IOPD) patients + healthy donors	Fibrin + Matrigel hydrogel in custom PDMS molds	GAA-deficiency, lysosome enlargement, glycogen accumulation.	rhGAA enzyme or AAV-mediated GAA partially reduced glycogen; incomplete rescue of functional deficits.	Partial disease features, no additional cell types.	[109]
Sarcopenia/aging	Primary human myoblasts from young and aged donors	Collagen I–Matrigel hydrogel incorporated in a PDMS-based microfluidic chip	Lower fusion index (36% vs. 69% to young muscle); Impaired and non-sychronous contraction	N/A	lack of additional cell types to mimic full muscle complexity	[114]
	Primary mouse myoblasts from young (3–5 mo) and aged (21–23 mo) mice	Fibrin/Matrigel hydrogel cast in a PDMS chamber	Recapitulated atrophy, reduced contractile function, and delayed regeneration	Supplementation with recombinant α-klotho, extracellular vesicles (EVs), and C4b inhibition via CFI; improved myotube regeneration and enhanced force.	Mouse cells; lacks multicellular complexity	[115]
	Primary human myoblasts from skeletal muscle biopsies	Bovine fibrinogen and Matrigel cast in PDMS molds	• Disease phenotype induced by chronic exposure of differentiated EMTs to IFN-γ (20 ng/mL for 7 days) • Key features: myofiber atrophy, reduced contractile strength, slower kinetics, impaired calcium transients, and disrupted sarcomere organization	Both electric stimulation (exercise) and JAK inhibitors (tofacitinib or baricitinib) prevented IFN-γ–induced muscle deficits; "exercise" partially restored force and structure while JAK inhibitors fully normalized contractile and calcium-handling properties	The chronic IFN-γ model may not capture the full complexity of in vivo inflammation; limited donor diversity	[116]
Obesity related metabolic syndrome	C2C12 + 3T3-L1 adipocytes	Bovine collagen I	• Direct or indirect co-culture C2C12 based 3D engineered muscle with 3T3-L1 adipocytes • Recapitulated metabolic crosstalk (higher lipolysis, lipid accumulation, insulin resistance insights)	No rescue; tested Isoproterenol (β-adrenergic agonist) response	Non-human cell lines; No contractility assessment; lack vascular/immune cell types in 3D muscle	[93]
	Primary rat myoblasts + Microvascular fragments (MVFs) from adipose tissue	Fibrin hydrogel (formed by mixing fibrinogen with thrombin) cast in PDMS molds	• Diabetic-like muscle pathology characterized by increased intramuscular adipose tissue (IMAT) • Co-culture showed disorganized myotube and vessel alignment, impaired mechanical integrity, and altered insulin responsiveness	N/A	Lack of active contractile (twitch/tetanus) measurements; in vitro adipogenic induction may not fully recapitulate human pathology; use of rat cells may limit direct translational relevance	[117]
	C2C12	Composite hydrogel of 1% alginate and 1 mg/mL collagen type I	Insulin resistance induced by treating differentiated 3D myotubes with palmitate (PA; optimal at 0.1 mM)	Pharmacological treatment using known insulin-sensitizers (rosiglitazone and metformin) plus screening of 165 candidate compounds	Focuses solely on metabolic endpoints (glucose uptake) without contractile assessment; uses a murine cell line (C2C12) that may not fully recapitulate human skeletal muscle physiology	[118]
	C2C12	65% Type I rat tail collagen, 20% Matrigel, and 10% 10X MEM, cast into 3D printed inserts	• 3D constructs exposed to exogenous fatty acids (OPLA mixture: oleic, palmitic, linoleic, α-linoleic acids at 45:30:24:1% ratio) at 200 µM and 800 µM for 4 days • Exhibits lipid droplet accumulation, altered lipid metabolism gene expression, reduced myosin heavy chain coverage, and decreased force generation	N/A	Short-term FA exposure may not mimic chronic conditions; lacks additional cell types to mimic the full in vivo microenvironment	[119]

## Data Availability

Not applicable.

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
