# Peer review of "From 2D Myotube Cultures to 3D Engineered Skeletal Muscle Constructs: A Comprehensive Review of In Vitro Skeletal Muscle Models and Disease Modeling Applications"

_cells, 2025, doi:10.3390/cells14120882_

Round 1
Reviewer 1 Report
Comments and Suggestions for Authors
This present review is an important work for the field by combining relevant works related to the biology of the skeletal muscle cell. For this reason, I could not find any improvements needed and I strongly recommend it to be accepted for publication.
Author Response
"This present review is an important work for the field by combining relevant works related to the biology of the skeletal muscle cell. For this reason, I could not find any improvements needed and I strongly recommend it to be accepted for publication."
Response: Thank you for your thoughtful evaluation and supportive recommendation. We greatly appreciate your time considering our work.
Reviewer 2 Report
Comments and Suggestions for Authors
The review article submitted by Dr. Cao and Dr. Warren is very interesting and well-structured and provides useful information, both technical and conceptual to a broad readership. I suggest to add a few interesting mouse models of Emey-Dreifuss Muscular Dystrophy.
Author Response
Comment: The review article submitted by Dr. Cao and Dr. Warren is very interesting and well-structured and provides useful information, both technical and conceptual to a broad readership. I suggest to add a few interesting mouse models of Emey-Dreifuss Muscular Dystrophy.
Response: We sincerely appreciate the reviewer’s thoughtful feedback and their recognition of the manuscript’s value to the field. The suggestion to incorporate mouse models of Emery-Dreifuss muscular dystrophy (EDMD) is well-noted. However, as this review focuses specifically on 3D engineered muscle tissue (EMT) models for studying muscular disorders, we have prioritized expanding Section 3.1 to include recent EMT models that recapitulate EDMD pathology, rather than in vivo mouse models.
Reviewer 3 Report
Comments and Suggestions for Authors
In the present review, Cao and Warren retrace key studies highlighting the advantages of transitioning from standard 2D muscle cultures to more complex 3D bioengineered constructs that better recapitulate muscle complexity. While the review is well-written and likely to be of broad interest to the field, certain sections feel somewhat superficial and would benefit from being improved in term of content and organized into images or tables for improved clarity.
The purpose of Figure 1 is not entirely clear. The review primarily focuses on muscle tissue engineering rather than detailing niche composition or general muscle structure; therefore, Figure 1 appears somewhat out of context. I kindly encourage the authors to revise this figure and consider adding an additional illustration that outlines a general workflow for successfully transitioning from 2D to 3D muscle cultures. Additionally, a table summarizing the biomaterials and scaffold types used—along with their corresponding references—could significantly enhance the impact of the review and improve readability for some of the more technical sections.
It would be valuable mentioning the real future challenge of muscle tissue engineering, for modeling with success human muscle diseases. At first, the goal is to create at first a miniaturized model that faithfully recapitulate the muscle niche taking into account the coordinated participation of all muscle cell types, including FAPs, endothelial cells, immune cells and etc. Which are the current limits for reaching this ambition task?
The discussion of synthetic hydrogels is rather brief, despite a substantial body of literature focusing on their use. For example, several groups have demonstrated the effectiveness of PEG-fibrinogen, in combination with myogenic progenitors, for reconstructing functional muscle-mimetic constructs capable of repairing volumetric muscle loss (VML) injuries in mice. Including this important literature seems essential to fully cover the topic of the review.
Finally, the section on current challenges and future directions should be strengthened by mentioning the need to develop supportive approaches that enable molecular-level monitoring of cell behavior in 2D versus 3D environments. This is a crucial step for scaling up and translating years of research based on standard culture systems. For instance, Reggio et al. recently used high-resolution proteomics to demonstrate that enhanced differentiation of muscle progenitor cells was achieved by creating a 3D, low-mitogenic environment that promotes differentiation. Highlighting such studies would provide valuable context and depth to this section.
Author Response
Comment 1: The purpose of Figure 1 is not entirely clear. The review primarily focuses on muscle tissue engineering rather than detailing niche composition or general muscle structure; therefore, Figure 1 appears somewhat out of context. I kindly encourage the authors to revise this figure and consider adding an additional illustration that outlines a general workflow for successfully transitioning from 2D to 3D muscle cultures. Additionally, a table summarizing the biomaterials and scaffold types used—along with their corresponding references—could significantly enhance the impact of the review and improve readability for some of the more technical sections.
Response: Thanks for helping us to improve our review manuscript. In response, we have completely redesigned Figure 1 to present a clear workflow for transitioning from 2D myotube cultures to 3D engineered muscle constructs for better fit for this review. Additionally, we have updated table 1 summarizing the key biomaterials and scaffold designs used in representative in vitro skeletal muscle studies, each accompanied by its corresponding reference, to improve technical clarity and reader accessibility.
Comment 2: It would be valuable mentioning the real future challenge of muscle tissue engineering, for modeling with success human muscle diseases. At first, the goal is to create at first a miniaturized model that faithfully recapitulate the muscle niche taking into account the coordinated participation of all muscle cell types, including FAPs, endothelial cells, immune cells and etc. Which are the current limits for reaching this ambition task?
Response: Thanks for pointing this out and guiding us to discuss the current challenges and future directions in greater depth. We have added a new paragraph in section 4.1 (Line 589-603) to discuss multicellular integration into the EMTs and the current key limitations. This addition significantly strengthens our discussion on future directions by outlining the specific hurdles that must be overcome to achieve truly biomimetic human muscle disease models.
Comment 3: The discussion of synthetic hydrogels is rather brief, despite a substantial body of literature focusing on their use. For example, several groups have demonstrated the effectiveness of PEG-fibrinogen, in combination with myogenic progenitors, for reconstructing functional muscle-mimetic constructs capable of repairing volumetric muscle loss (VML) injuries in mice. Including this important literature seems essential to fully cover the topic of the review.
Response: Accordingly, Section 2.1 (Line 241-257) has been significantly strengthened with focused discussion on synthetic hydrogels, including PEG-fibrinogen systems, highlighting their critical role in volumetric muscle loss (VML) repair in restoring contractile force in murine VML models.
Comment 4: Finally, the section on current challenges and future directions should be strengthened by mentioning the need to develop supportive approaches that enable molecular-level monitoring of cell behavior in 2D versus 3D environments. This is a crucial step for scaling up and translating years of research based on standard culture systems. For instance, Reggio et al. recently used high-resolution proteomics to demonstrate that enhanced differentiation of muscle progenitor cells was achieved by creating a 3D, low-mitogenic environment that promotes differentiation. Highlighting such studies would provide valuable context and depth to this section.
Response: Agree. We have accordingly strengthened the Future Directions section (Line 634-646) by adding discussion on advanced profiling techniques essential for scaling engineered muscle platforms. We explicitly emphasize the value of proteomics for mapping differentiation pathways across dimensions, citing Reggio et al.'s recent work demonstrating how a low-mitogenic 3D environment suppressed proliferative signaling while enhancing myogenic maturation, revealing mechanistic advantages over 2D systems. This addition highlights how molecular phenotyping bridges fundamental research with clinical translation.
Reviewer 4 Report
Comments and Suggestions for Authors
The review by Cao and Warren covers a very interesting and important topic. The manuscript is clearly structured, well-written and consistently understandable, reflecting the current state of knowledge on 3D cell culture models. While reading, I noticed a few typos, which can be quickly corrected in the proof:
- line 44: Better write ‘movement of organisms’ instead of ‘organismal movement’
- line 71: Add full stop between ‘capacity’ and ‘In’
- line 175: Delete the superfluous gap between ‘It controls’
- line 225: For instance, Perniconi et al. in italic?
- line 298: Add full stop between ‘[40]’ and ‘Such’
- line 575: Add gap between necrosis[117]
- many sites: search systematically and delete superfluous gaps between words (e.g. lines 341, 344, 358)
Author Response
Comments: The review by Cao and Warren covers a very interesting and important topic. The manuscript is clearly structured, well-written and consistently understandable, reflecting the current state of knowledge on 3D cell culture models. While reading, I noticed a few typos, which can be quickly corrected in the proof:
- line 44: Better write ‘movement of organisms’ instead of ‘organismal movement’
- line 71: Add full stop between ‘capacity’ and ‘In’
- line 175: Delete the superfluous gap between ‘It controls’
- line 225: For instance, Perniconi et al. in italic?
- line 298: Add full stop between ‘[40]’ and ‘Such’
- line 575: Add gap between necrosis[117]
- many sites: search systematically and delete superfluous gaps between words (e.g. lines 341, 344, 358)
Response: Thanks for your time considering our work and for correcting the typos. We have revised all these typos mentioned above in-text as suggested.
Round 2
Reviewer 3 Report
Comments and Suggestions for Authors
In this present form the manuscript I suitable for publication